# Hierarchical organization of context in the hippocampal episodic code

**Susumu Takahashi**[1,2,3]*

[1]Laboratory of Neural Circuitry, Graduate School of Brain Science, Doshisha University, Kizugawa, Japan; [2]Precursory Research for Embryonic Science and Technology (PRESTO), Japan Science and Technology Agency, Kawaguchi, Japan; [3]Faculty of Computer Science and Engineering, Kyoto Sangyo University, Kyoto, Japan

**Abstract** The hippocampal system appears to be critically important in establishing episodic memory of both internal and external events within contexts as well as spatial memory, which enables flexible spatial navigation. However, the neuronal substrates that function across different memories in the hippocampal system are poorly understood. I monitored large-scale activity patterns of hippocampal neuronal ensembles in rats performing a novel, continuous task that combined one visually guided and two memory-guided types of navigations in a constant environment. I found that the activity patterns of the hippocampal ensemble represent spatiotemporal contexts (journeys) constructed by temporally ordered past, present and expected future places in tandem with visually or mnemonically guided non-spatial contexts (task-demands) to form episodes. This finding therefore suggests that the hierarchical organization of contexts based on pattern separation and completion enables the hippocampus to play a dual role in spatial navigation and recall of episodic memory.

*For correspondence: stakahas@mail.doshisha.ac.jp

**Competing interests:** The author declares that no competing interests exist

## Introduction

There have been two frameworks suggested for the principal role of the hippocampus in memory: spatial memory and episodic memory (*Scoville and Milner, 1957*; *O'Keefe and Nadel, 1978*; *Vargha-Khadem et al., 1997*; *Eichenbaum et al., 1999*). During spatial behaviors, hippocampal pyramidal neurons discharge specifically at a certain location in the environment (the 'place field'), suggesting that hippocampal neurons function in spatial memory (*O'Keefe and Dostrovsky, 1971*; *McNaughton and Morris, 1987*; *Muller et al., 1987*). In contrast, episodic memory provides a record of past experiences in relation to a broad range of materials (*Wood et al., 1999*) and is structured by spatial, temporal, and non-spatial contexts (*Tulving, 1983*). Recent studies indicate that on the basis of the pattern separation and completion proposed by Marr (*Marr, 1969*), the place field of a given neuron changes (remapping) relative to three different contexts: spatial context characterized by environmental features (*Bostock et al., 1991*; *Kentros et al., 1998*), spatiotemporal context (journey) affected by the experienced origin and expected destination (*Frank et al., 2000*; *Wood et al., 2000*), and internally and externally guided non-spatial context (task-demand) (*Gothard et al., 1996*; *Anderson and Jeffery, 2003*; *Smith and Mizumori, 2006*). Moreover, during a temporal gap between events, the place cells encode successive moments or fire at a specific part of an episode (*Pastalkova et al., 2008*; *MacDonald et al., 2011*). These lines of evidence suggest that the hippocampal place code is not only closely related to the 'where' of episodic memory but also to the 'when and how'. To test whether the changes and dynamics of the hippocampal place code are the neuronal basis of spatial and episodic-like memory, it is necessary to conduct a multifaceted experiment in which many hippocampal neurons are simultaneously monitored during repeated exposures to either spatiotemporal or non-spatial contexts in a constant spatial environment.

**eLife digest** A little over 10 years ago, researchers discovered that a brain region called the hippocampus is larger in London taxi drivers than it is in the general population. This tied in with results from animal studies, which had revealed a key role for the hippocampus in spatial navigation and memory. However, other work has shown that the hippocampus is equally important for remembering personal experiences—a form of memory known as episodic memory.

Many neurons in the hippocampus display 'place fields', which means that they fire bursts of action potentials whenever an animal is in a specific location. Place fields tend to remain stable during repeated visits to an environment: the same cells fire whenever the animal returns to a particular place. However, if the animal enters a new environment, a neuron might adopt a different place field or not show any place field at all. This phenomenon is known as remapping.

Now, Takahashi has provided further insight into the circumstances under which such remapping occurs. He recorded from large numbers of neurons in the rat hippocampus—in a subregion called CA1—as the animals moved through a maze shaped like a digital figure '8'. The rats had to perform three tasks within the maze: one guided by visual cues, and two that were memory-based.

In the visual task, a light informed the rats to turn either left or right to obtain a reward. In the first memory task, the rats had to alternate their choices to obtain the reward, running through the maze from right-to-left and then from left-to-right (non-delayed spatial alternation). The second memory task worked the same way, except that the rats had to wait 5 s before turning left or right (delayed spatial alternation).

Takahashi compared the responses of hippocampal CA1 neurons as rats performed the three tasks. As expected, he found that neurons tended to remap their place fields based on the animal's initial and final locations in the maze, regardless of which task the animal was performing. Surprisingly, however, neurons with specific place fields distinguished between the three tasks by firing at different rates in each.

By combining information about the locations and rates at which large assemblies of neurons fired, Takahashi found that he could accurately predict which task a rat had been performing, where it had come from, and where it had ended up, because the place field remapping was hierarchically organized. Moreover, the prediction could be made even before the rat had completed the task. Overall, these results add to our understanding of how the hippocampus performs its dual roles in spatial navigation and episodic memory.

## Results

### A continuous task with one visually guided and two memory-guided decisions

Rats were trained to navigate their way through a figure '8' maze in a continuous task that incorporated both visual discrimination and two types of memory-guided responses (see 'Materials and methods', *Figure 1*). The task was divided into three subtasks: visual discrimination (VD), non-delayed spatial alternation (SA) and delayed spatial alternation (DA). In the VD subtask, a cue light on the left or right side was illuminated to indicate which direction to turn at a decision point (DP, *Figure 1A*) to receive a reward (medial forebrain bundle [MFB] electrical stimulation). In both the SA and DA subtasks, cue lights on both sides were illuminated and the rats had to alternate between left and right at the decision point (*Figure 1B*). The difference between the SA and DA subtasks was that in the DA subtask, a barrier appeared at the middle of the central stem for 5 s to interpose a delay period. During the delay period, the rats paused steadily in the forward direction in front of the barrier. Some lines of evidence suggest that the SA-type task is often affected by hippocampal lesions (*O'Keefe and Nadel, 1978*). By contrast, a recent study implied that the SA-type task is independent on the hippocampus (*Ainge et al., 2007b*). Unlike the visually cued VD and procedural memory-guided SA subtasks, the episodic-like memory-guided DA type of task is critically dependent on the hippocampus (*Ainge et al., 2007b*; *Ferbinteanu et al., 2011*). The rats continuously performed these subtasks at least twice in a task over the course of approximately 1 hr (*Figure 1D*). This task allowed for examinations

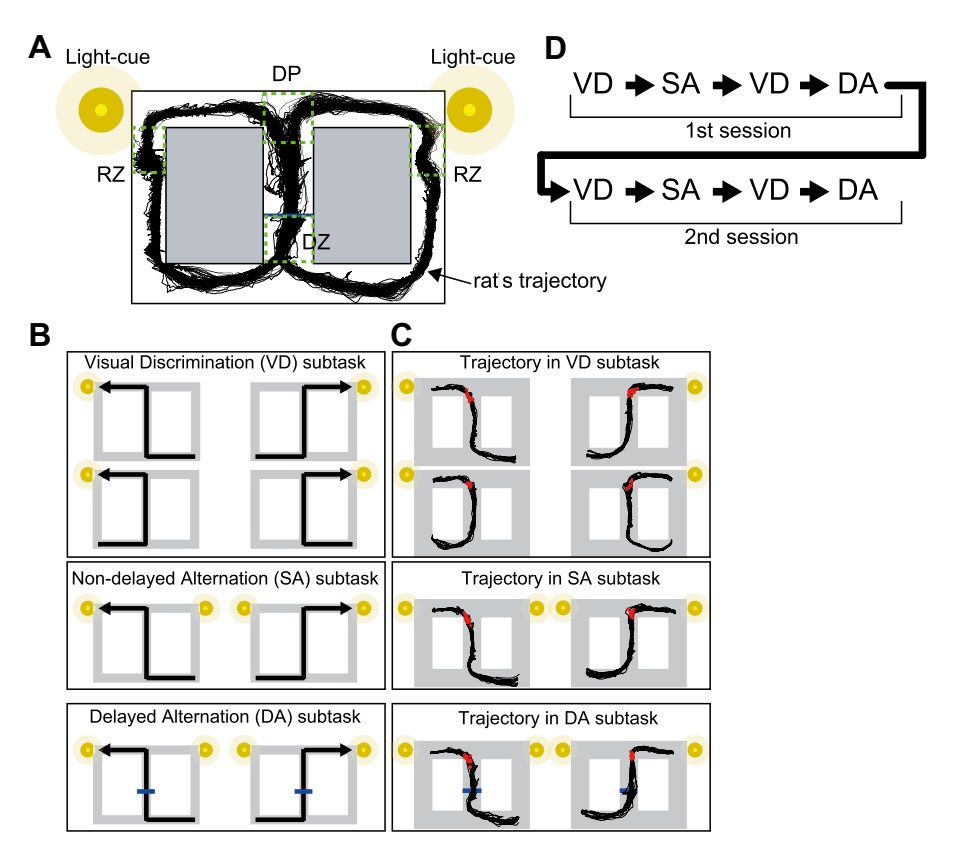

**Figure 1**. Task configuration. (**A**) The rat has to decide which direction to turn at the decision point (DP). If correct, the rat gets reward signals (MFB stimulations) at one of the reward zones (RZ). In the DA subtask, a barrier appears for 5 s in the central stem (blue line) so that the rat has to wait in the delay zone (DZ). An example of a running trajectory of a rat (G111125) is superimposed on the maze. (**B**) Possible journeys for three subtasks. For the VD subtask, one of the light cues is illuminated to indicate a direction. For the memory-guided (SA and DA) subtasks, both of the light cues are illuminated to alternate the direction based on memory. (**C**) Examples of running trajectories of the rat (G111125). The red dots indicate the location of the onset of the estimated turn for each lap. (**D**) The subtask sequence in a task. Each subtask consisted of 20 laps, except the second VD subtask in each session, which only consisted of 10 laps.

of the influence of internal and external events on hippocampal neuronal activity and hippocampus-dependence both between and within different journeys and task-demands.

## Coding of place, journey and task-demand in the hippocampal place cells

Ensemble activity was recorded from a total of 1119 pyramidal cells in the CA1 of the dorsal hippocampus using arrays of 10 extracellular dodecatrodes (*Takahashi and Sakurai, 2005*) in five rats running in the maze. All rats were familiarized with the task before recording, and performed the task with a high level of accuracy (overall: 99 ± 2%, VD: 99 ± 1%, SA: 97 ± 3%, DA: 99 ± 1% correctly performed [mean ± SEM]; *Table 1*). Laps containing errors of cue interpretation were excluded from the following analyses.

Several studies suggest that the hippocampal place code is influenced by running speed and head direction (*McNaughton et al., 1983*; *Wiener et al., 1989*; *Muller et al., 1994*; *Czurko et al., 1999*). In addition, even small differences in location on a maze can strongly influence firing rates (*Muller et al., 1994*). Although in the present work, these factors were considered to be minimized in the central stem of the maze because the running speed, head direction, and lateral position appeared highly consistent, an analysis of the place cells was employed between different journeys within a given subtask or between different task-demands within a journey in order to quantitatively account for any

**Table 1.** Variability between rats: behavioral and electrophysiological measurements

| Rat No. | B120303 | F120112 | G111125 | K120224 | L120221 | Total |
|---|---|---|---|---|---|---|
| No. of erroneous laps | 0 | 1 | 0 | 5 | 1 | |
| Errors (%) | 0 | 0.7 | 0 | 3.5 | 0.7 | |
| Running speed (cm/sec ± SD) in the central stem | 33.13 ± 6.17 | 45.61 ± 11.73 | 22.09 ± 15.08 | 41.77 ± 11.77 | 52.03 ± 16.95 | |
| No. of pyramidal cells | 238 | 74 | 412 | 193 | 202 | 1119 |
| No. of place cells (spatial information > 1.0 bits/spike) | 127 | 21 | 199 | 59 | 116 | 522 |
| Spatial information (bits/spike, mean ± SEM) | 2.39 ± 0.21 | 1.26 ± 0.36 | 2.47 ± 0.21 | 1.21 ± 0.18 | 2.50 ± 0.24 | |
| Unit isolation quality (isolation distance, mean ± SEM) | 22.42 ± 1.70 | 27.69 ± 2.81 | 45.41 ± 7.86 | 23.26 ± 1.59 | 33.60 ± 2.11 | |
| No. of interneurons | 19 | 5 | 24 | 15 | 10 | 73 |

remaining effects of these potentially confounding factors. In this analysis, for each place cell in which the spatial information in at least one trial type was above 1.0 bits/spike, differences in firing rates associated with those differences in the central stem were analyzed using a one-way ANCOVA. Running speed, head direction and lateral position were included as covariates in the ANCOVA model. Firing rates of over 268 of the 522 place cells in the central stem differed significantly between different journeys within a given subtask or between different task-demands within a journey ($p < 0.05$; *Figure 2*). To quantitatively examine the remapping of place fields without influences of the running speed, head direction, and lateral position, only place cells which fired at significantly different rates in the central stem on different journeys within a given subtask or different task-demands within a journey, even when all of these factors were taken into account, were examined in the following analyses.

The place codes could be changed in terms of firing rates (rate remapping) or both firing rates and location (global/complete remapping; *Bostock et al., 1991*; *Leutgeb et al., 2005*). For each place cell, I independently compared the firing location and rate of the place fields during task performance. The spatial similarity was measured by calculating the spatial correlation between a pair of maps. The rate similarity was measured as one minus the normalized difference between the firing rates. Thus, the spatial similarity measurement was independent of rate similarity. In an identical trial type (i.e., combination of journey and task-demand), the distributions of the measurements between first and second exposures (i.e., first vs second exposure in the right-to-left or left-to-right journey in the VD, SA or DA subtasks) were highly similar (median Spearman correlations [$r_s$] > 0.67; median rate similarity indices [$r_r$] > 0.80; *Figures 3A, 4B,C*), suggesting that, as in numerous previous studies on the hippocampal place code (*O'Keefe and Dostrovsky, 1971*; *McNaughton and Morris, 1987*; *Muller et al., 1987*), the pyramidal cells are place-specific during identical visually or mnemonically guided behaviors in a constant environment. Within a given task-demand, the distributions of the spatial correlation and the change in firing rates in the place fields in the central stem between different journeys (i.e., a right-to-left vs left-to-right journey in the VD, SA, or DA subtasks) were significantly lower than those seen under control conditions (repeated exposures to identical trial types) (median $r_s$ < 0.45, $Z$ < −22.04, $p < 0.001$; median $r_r$ < 0.68, $Z$ < −8.96, $p < 0.001$, Wilcoxon rank sum test; *Figures 3B and 4B,C*), suggesting that place-specific activity is strongly dependent on the journey (*Frank et al., 2000*; *Wood et al., 2000*; *Ferbinteanu and Shapiro, 2003*; *Bower et al., 2005*; *Ainge et al., 2007a*). In contrast, in an identical journey, the distributions of the spatial correlations among the VD, SA and DA subtasks (i.e., VD vs SA subtask, VD vs DA subtask, SA vs DA subtask in the right-to-left or left-to-right journey) were similar or higher than under control conditions (median $r_s$ > 0.75, $Z$ > 1.07, n.s. or $p < 0.001$, Wilcoxon rank sum test; *Figures 3B and 4B*); however, the distributions of the change in firing rates in the place fields were clearly lower (median $r_r$ < 0.80, $Z$ < −2.79, $p < 0.01$, Wilcoxon rank sum test; *Figures 3B and 4C*), indicating that only the rate of a place-specific activity is modulated with a given task-demand (*Smith and Mizumori, 2006*). Taken together, the results suggest that irrespective of hippocampus-dependence (*Morris and Frey, 1997*; *Ainge et al., 2007b*; *Ferbinteanu et al., 2011*), the hippocampal pyramidal cells in the maze were not only activated by a specific location, but

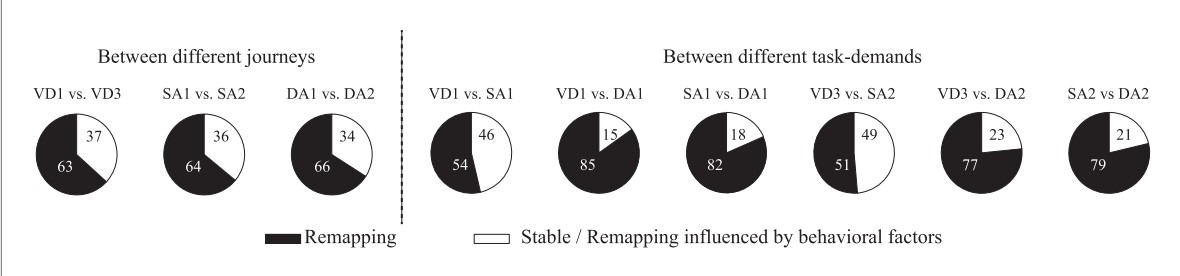

**Figure 2**. Proportion of differentially firing place cells. The proportion of place cells showing significantly different firing rates between different journeys within a given subtask and between different task-demands within a journey when running speed, head direction and lateral position were taken into account. Numbers indicate percentages.

also by the experienced past and expected future places accompanied by the visually guided or memory-guided task-demands (see *Figure 3—figure supplement 1*).

## Episode-specific neuronal trajectory of the hippocampal ensemble activity

The activity patterns relative to the place, journey and task-demand during tasks suggested that the dynamics of the ensemble activity pattern rather than just the firing pattern of single cells should be considered. The dynamics of neuronal ensemble activity as a trajectory through a state space were therefore analyzed (neuronal trajectory; see 'Materials and methods'; *Figure 5*) (*Briggman et al., 2005*; *Mazor and Laurent, 2005*; *Harvey et al., 2012*). At each specific location, the activity state of the ensemble containing *n* simultaneously recorded neurons was defined as a point in an *n*-dimensional space, with each dimension representing the activity of a single neuron. In the trajectory of a journey within a given subtask, the neuronal ensemble activity tended to move in similar orbits that started and ended at similar positions. Moreover, the trajectories of different journeys exhibited unique orbits (*Figure 6A,B*). To quantify the trajectory specificity, I used a classifier based on the distance from an individual lap trajectory to the mean right-to-left/left-to-right trajectories at single locations. In each subtask, the neuronal trajectory, consisting of all of the simultaneously recorded place cells, was sufficient to separate the left-to-right journey from the right-to-left journey with high accuracy, and vice versa (*Figure 6C*; p<0.001, binomial test, chance = 0.5). Surprisingly, even in visually guided situations in which memory-guided decisions could not be utilized (i.e., left-to-left/right-to-right journey in the VD subtask), the accuracy was sufficiently better than chance levels before the rat behaviorally chose its next direction (prospective coding; *Figure 6D*; p<0.001, chance = 0.5; turn onset: *Figure 1C*). Furthermore, this trajectory could be used to predict all possible trial types (i.e., eight possibilities; VD1, VD2, VD3, VD4, SA1, SA2, DA1 and DA2) on single laps at better than chance levels even during periods of running through the central stem of the maze in which the head direction, speed and place factors were almost constant for each rat (*Figure 7*, p<0.001, binomial test, chance = 0.125; *Table 1*). Thus, the activity in the hippocampus could be considered to be divergent, episode-specific trajectories through a state space of neuronal ensemble activity (see *Figure 7—figure supplement 1*).

## Relative ability for distinguishing journeys, task-demands, and trial types in the hippocampal episodic code

The above results suggest that hippocampal ensemble activity contains sufficient information concerning journey, task-demand and trial type. To compare the relative information content among them for neuronal ensemble activity of each rat, I quantitatively assessed the performances of the binary classification of neuronal trajectories in the central stem of the maze between a pair of journeys irrespective of subtask, between a pair of task-demands irrespective of journey, and between a pair of trial types. For all categories, the average output of the binary classifiers showed a high level of accuracy at the ensemble level (*Figure 8*, p <0.001, binomial test, chance = 0.5) and no significant differences among them (p >0.05, Kruskal–Wallis test with Turkey honestly significant difference [HSD] post-hoc test). The results suggest that the abilities of the hippocampal ensemble to identify journeys, task-demands, and trial types are sufficiently high and relatively similar to each other.

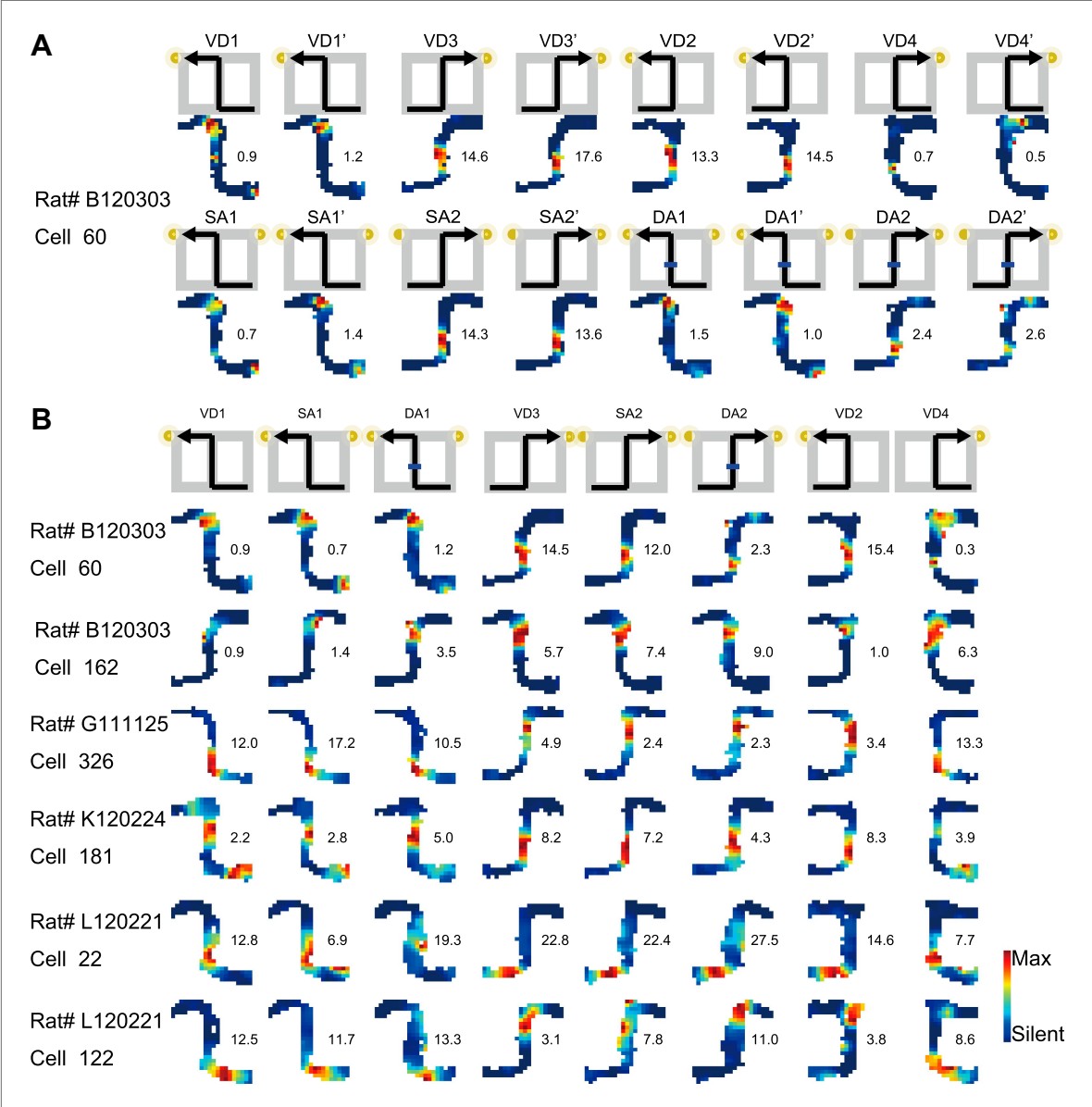

**Figure 3**. Journey-dependent coding of a place-specific activity modulated with task-demand. Representative color-coded rate maps for eight possible trial types. (**A**) The rate maps with the first cell in (**B**) show stable place-specific activity in repeated sessions of an identical journey within a given task-demand. The rate was coded on a color scale from blue (silent) to red (maximum rate). The pixels that were not sampled are white. The symbols above the map indicate the journey, task-demand and illuminated light-cue. For each cell, the rate scale corresponds to the peak firing rate in Hz in that condition (indicated to the right of the rate maps). (**B**) The rate maps for six CA1 pyramidal cells. The maps are averages for repeated sessions. Each row shows the data from one pyramidal cell. Note that different journeys in the same place resulted in place fields similar in locations, but different task-demands caused different firing intensities in those place fields.

The following figure supplements are available for figure 3:

**Figure supplement 1**. Schematic of the main finding with regard to place fields.

## Hierarchical representation of journey and task-demand in hippocampal ensemble activity

The results obtained showing the similarities of the place fields may imply the pattern completion/generalization of the journey represented in the hippocampal ensemble activity in the spatial domain. If this assumption is correct, knowing the neuronal trajectories for right-to-left and left-to-right journeys in one subtask

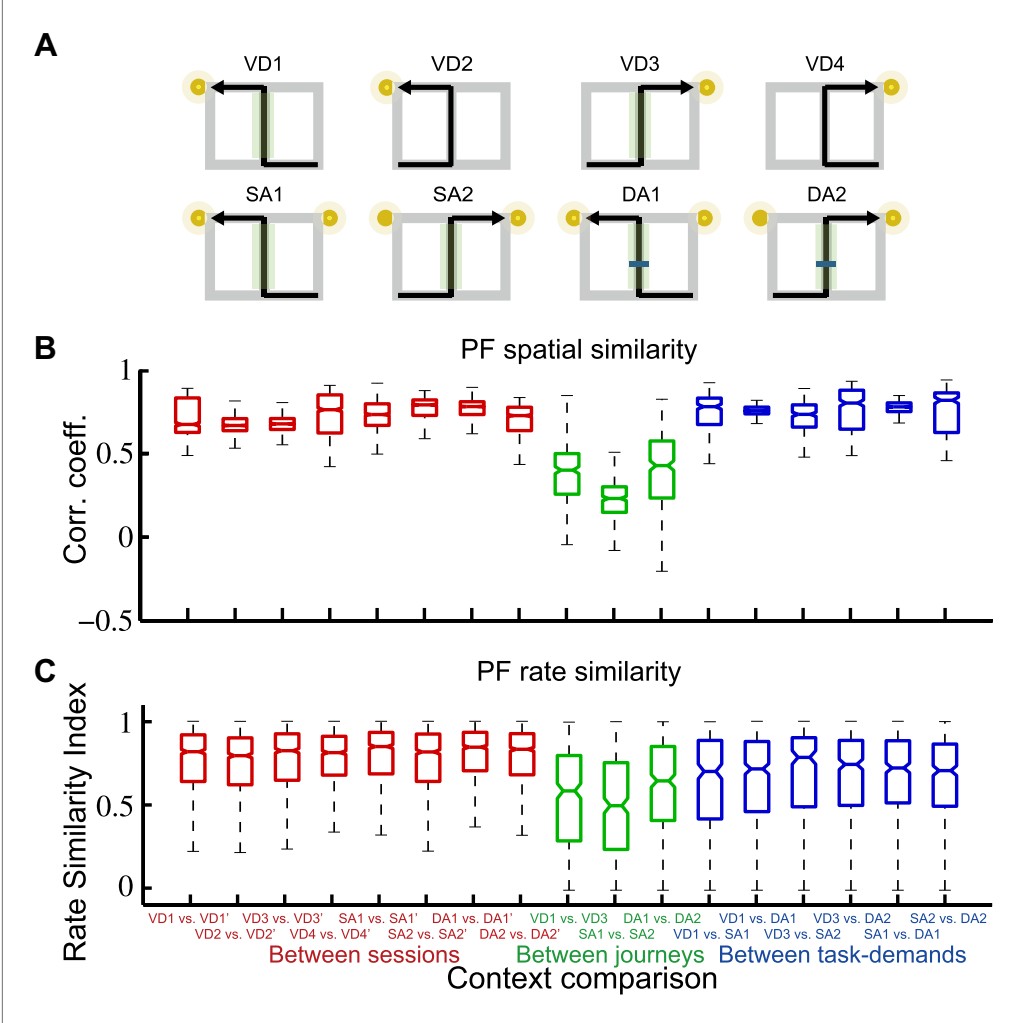

**Figure 4**. A quantitative assessment of the spatial and rate similarities between a pair of trial types. (**A**) The similarities were measured between a pair of trial types (VD1, VD2, VD3, VD4, SA1, SA2, DA1, and DA2) or between repeated exposures to an identical trial type. Because the central stem of the maze was the common running route among the different journeys within a given subtask, the region of interest (ROI) was set at the region highlighted in the green shaded box. Box plots of the spatial (**B**) and rate (**C**) similarities between repeated exposures to an identical journey and task-demand (red), between different journeys within a given task-demand (green), and between different task-demands within a journey (blue) (median, first and third quartiles, minimum, and maximum indicated). Repeated exposures are marked with an apostrophe.

should allow classification into right-to-left and left-to-right journeys in the other subtasks. To test this, the similarities between the ensemble activity patterns in the central stem of the maze were quantitatively analyzed using a distance-dependent classification scheme. As expected, based on the mean right-to-left/left-to-right trajectories in any one of three subtasks, the classifier outputs could predict the corresponding individual journeys within the other remaining two subtasks at a high level of accuracy at the ensemble level (see 'Materials and methods'; *Figure 9*, p<0.001, binomial test, chance = 0.5, n = 128 cells). These results suggest that hippocampal ensemble activity generalizes a journey over the given task-demands.

## Discussion

To summarize, although the same prominent environmental features were encountered in the same locations, the spatial representation of the hippocampal CA1 at various points in the maze were dissimilar in both firing location and rate among different journeys, irrespective of either visually or

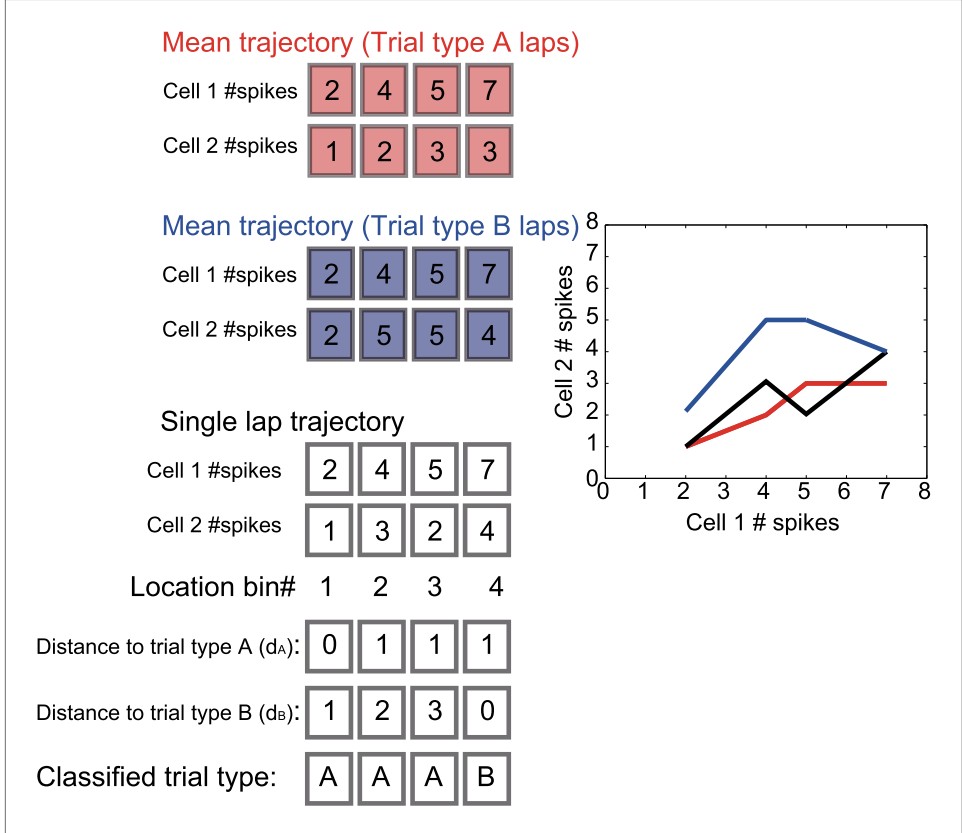

**Figure 5**. A schematic of a classification scheme based on the distances to the mean trajectories. At each location, the Euclidian distance from the trajectory on a single lap (black) to the mean trial type trajectories (red and blue) was calculated ($d_A$, $d_B$). If $d_A$ was less than $d_B$, the lap at the location bin was classified as trial type A, and vice versa for trial type B.

mnemonically guided demands, in a global remapping manner. The result supports the view that hippocampal pyramidal cells show journey-dependent coding during internally or externally guided goal-directed behaviors (*Ferbinteanu et al., 2011*). However, when the rats experienced changes in prominent, non-spatial features with internal and external events (i.e., subtask differences), the hippocampus only modulated the journey-dependent activity by primarily changing the intensity of the firing rates. The latter finding indicates that the extent to which the place fields remap is demand-specific (*Smith and Mizumori, 2006*) in a rate remapping manner. Neither the journey-dependent nor demand-specific coding was chiefly influenced by the running speed, head direction, or lateral position. In addition to the evidence from the place field measurements, even when the given task-demands were different, the classifier could sufficiently predict the journey from the ensemble activity patterns, suggesting that the journey representation is generalized and that non-spatial demand-specific representation is hierarchically ranked at a lower level. The neuronal trajectory of the ensemble activity differed enough to classify any trial type; prior to its behavioral choice, significant accuracy could be achieved even in a visually-guided situation in which memory-guided decision could not be made. The neuronal ensemble code for distinguishing trial types in the hippocampal ensemble activity was relatively equal to that for journeys and task-demands. Thus, the results imply that based not only on externally induced but also internally generated events, the hippocampus processes the combination of journey and task-demand as a single episode.

Previous studies have reported that hippocampal neurons might convey information related to where an animal is located (place-specific coding) (*O'Keefe and Dostrovsky, 1971*; *McNaughton and Morris, 1987*; *Muller et al., 1987*; *Bostock et al., 1991*; *Kentros et al., 1998*), where it had just been (retrospective coding) (*Frank et al., 2000*; *Wood et al., 2000*; *Ferbinteanu and Shapiro, 2003*;

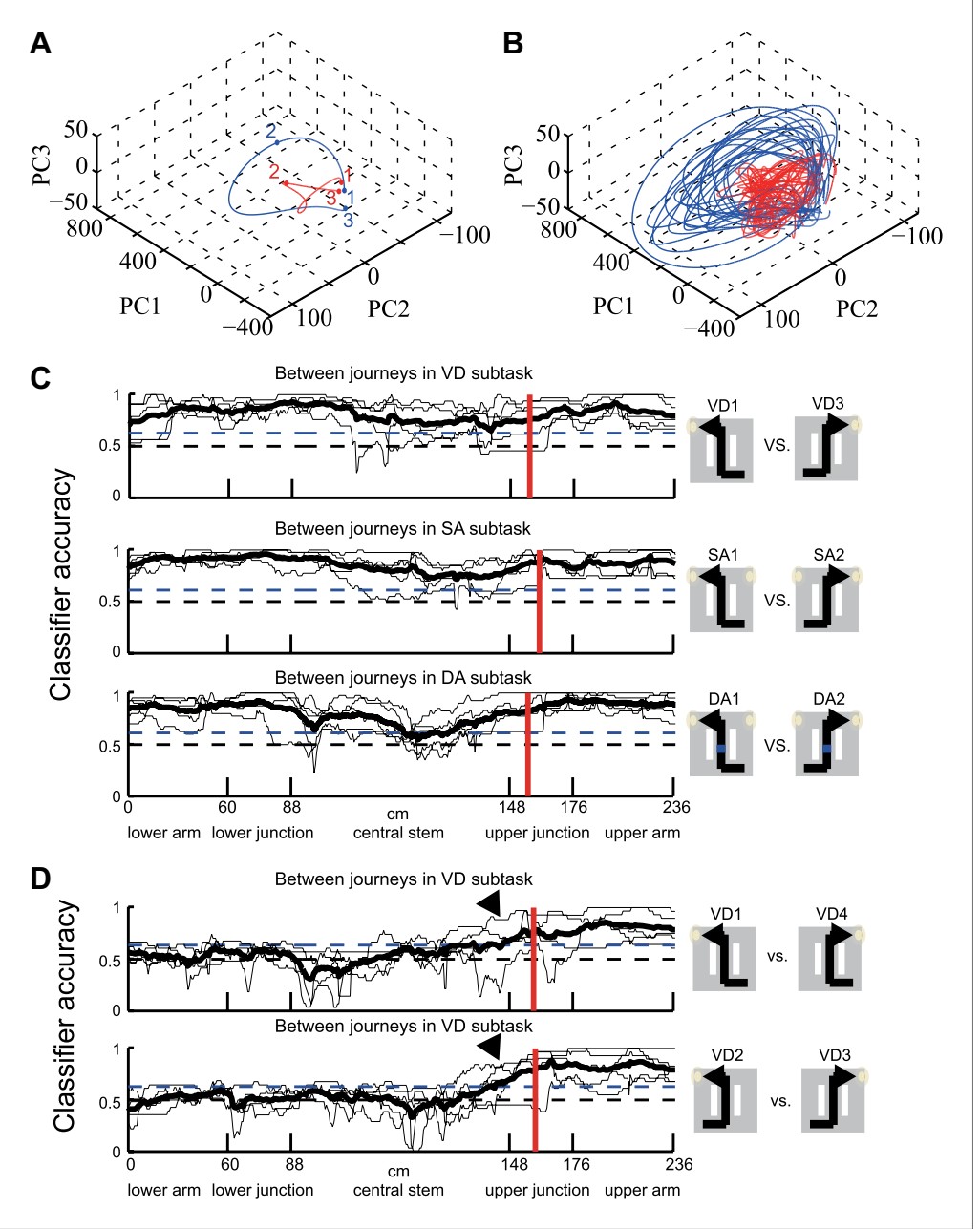

**Figure 6**. Neuronal trajectory of different journeys and different future directions. (**A**) An example time course of mean, journey-specific trajectories for right-to-left (red) and left-to-right (blue) laps in the SA subtask, plotted on the first to third principal component space (Rat #: B120303, n = 91 cells). The points marked 1, 2 and 3 correspond to the mean locations where the rat entered the lower arm, and turned and entered the reward zone, respectively. (**B**) Example individual trajectories for laps in the subtask in (**A**). (**C**) The classification accuracy of determining the journeys (shown in right) at different locations in the VD, SA, and DA subtasks (thin black line, n = 5 rats; solid black line, mean; blue dashed line, p=0.001, binomial test; black dashed line, chance level; red vertical line, mean turn onset). The classifier was based on a distance-dependent classification scheme. (**D**) As (**C**), except that the subtask condition under which the future direction is not predictable until arriving at the decision point. A high level of accuracy can be observed even before the onset of the turn (arrow heads).

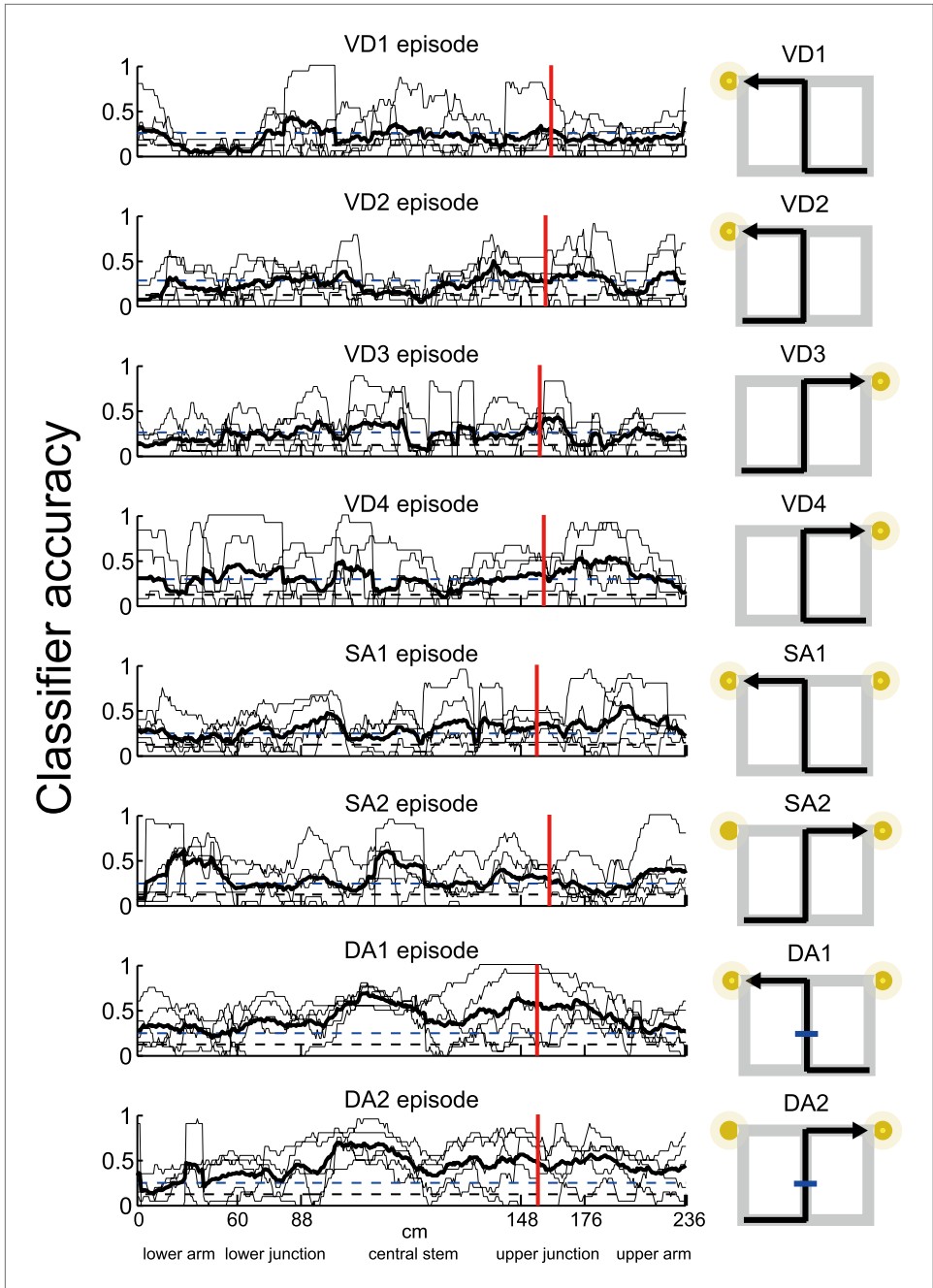

**Figure 7**. Episode-specific neuronal trajectories. The classification accuracy for determining a trial type from all of the possible trial types (episodes) (thin line, n = 5 rats; thick black line, mean; blue dashed line, p=0.001, binomial test; black dashed line, chance level). The classifier was based on a distance-dependent classification scheme. The average accuracy for each rat was significantly higher than chance at the respective specific locations.

The following figure supplements are available for figure 7:

**Figure supplement 1**. Schematic of the main finding with regard to episode-specific neuronal trajectories.

*Ferbinteanu et al., 2011*), where it is about to go next (prospective coding) (*Frank et al., 2000*; *Wood et al., 2000*; *Ferbinteanu and Shapiro, 2003*; *Ainge et al., 2007a*; *Ferbinteanu et al., 2011*) and what demand it currently faces (demand-specific coding) (*Gothard et al., 1996*; *Anderson and Jeffery, 2003*; *Smith and Mizumori, 2006*). However, the relationship among them in the

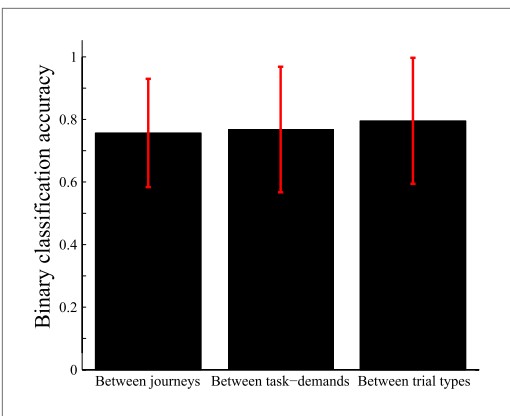

**Figure 8**. Comparison of accuracy of distinguishing journeys, task-demands and trial types from neuronal ensemble activity. Average binary classification accuracies of distinguishing pairs of journeys (i.e., VD1, SA1, and DA1 vs VD3, SA2, and DA2), pairs of task-demands (i.e., VD vs SA, VD vs DA, and SA vs DA), and pairs of trial types (i.e., all possible pairings of VD1, VD2, VD3, VD4, SA1, SA2, DA1, and DA2). For each rat, the accuracy was calculated based on the most frequently classified choices in the central stem. The error bars indicate SD.

hippocampal neuronal ensemble codes was unclear in these studies because each piece of information was independently examined. I report that all of this information can be simultaneously preserved in the hippocampal output to form episodes. The information for journey and task-demand is not horizontally but hierarchically organized; expected future events in the hippocampus can trigger episode separation. From the viewpoint of context, journey and task-demand can be considered to be spatiotemporal and non-spatial contexts, respectively. Thus, the hierarchical organization of spatiotemporal and non-spatial contexts in the hippocampal episodic code, which is based on the context version of pattern separation and completion through the remapping of place fields, not only enables flexible spatial navigation but also contributes to the recall of episodic memory.

## Materials and methods

### Animals
Five male Wister rats (300–400 g) were housed individually in cages (20 × 25 × 23 cm). All of them were implanted with a custom-made microdrive with 10 dodecatrodes and 2 reference electrodes. The rats were kept at 80% of free-feeding body weight. All rats were maintained on a 12-hr light/12-hr dark schedule. Testing occurred in the light phase.

### Surgery, electrode preparation and recording
Under isoflurane anesthesia, two stimulation electrodes were inserted into the MFB in the right lateral hypothalamus (AP 2.5, ML 1.0, DV 9.5); a custom-made microdrive with 10 independently movable 12-wire bundled dodecatrodes and 2 tetrodes were fixed to the skull above the left hippocampus (AP 3.8, ML 3.0, DV 0.5). Two tetrodes were used to record a reference signal. Each stimulation electrode consisted of two insulated stainless steel wires (coated diameter, 0.2 mm) together with ~0.5 mm of insulation removed from one tip. The 10 dodecatrodes were lowered into the hippocampal CA1 pyramidal layer within 2 weeks of surgery. In all cases, the unit-recording dodecatrodes and reference tetrodes were constructed from 8 μm and 12.5 μm HML-coated tungsten (99.95%) wires, respectively.

After the rats had made a full recovery (~2 weeks after surgery), they were allowed to explore the task maze and identify the reward zones (**Figure 1A**, RZ). During the physiological recording, the data from all channels of the dodecatrodes were unity-gain buffered, filtered (600 Hz–6 kHz), amplified (gain = 5000) and continuously sampled at 25 kHz on a 128-channel custom-made recording system with Lynx-8 amplifiers (Neuralynx, Bozeman, MT) and AD converters (16-bit resolution, National Instruments, Austin, TX). One channel of each dodecatrode was branched away, filtered (0.1 Hz–6 kHz) and amplified (gain = 500) to detect local field potentials (LFP). The recordings were included in the data analysis if sharp wave ripple events in the LFP in the corresponding dodecatrode were identified during the immobility period. Two sets of small red and green light-emitting diodes (10-cm separation), mounted above the headstage amplifier, were recorded by an overhead digital video camera and sampled at 60 Hz to track the rat's head position and direction. The neuronal data were recorded on a single recording day for each rat.

### MFB stimulation
After the surgery, the optimal stimulation parameters were obtained when the rat performed nose poke responses. The optimal MFB stimulation consisted of a train of 2 ms wide, 100–200 μA, biphasic current pulses, delivered at 100 Hz for 200 ms. In the correct laps, the optimal MFB simulation was

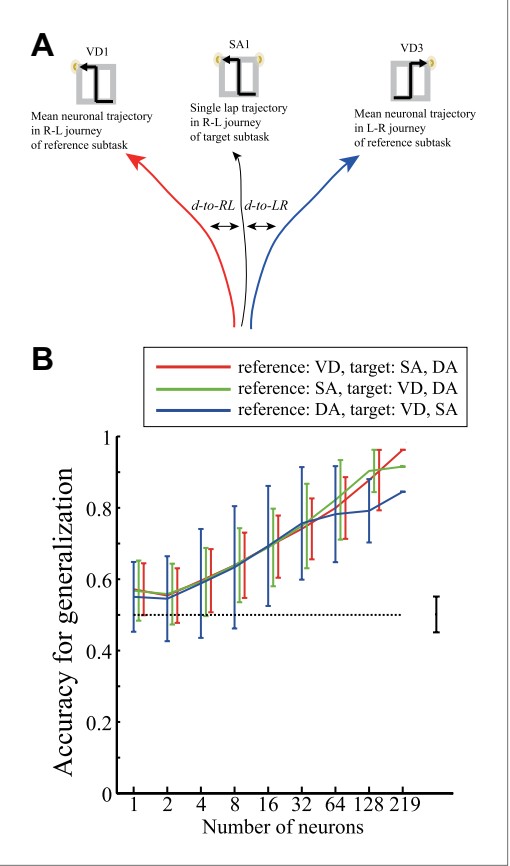

**Figure 9**. Generalization accuracy of determining a journey over task-demands from neuronal ensemble activity. (**A**) Schematic of a classification scheme for estimating generalization accuracy. At each location, the distance from the trajectory in two target subtasks on a single trial to the mean right-to-left (R-L) and left-to-right (L-R) journey trajectories in a reference subtask was measured (*d-to-RL* and *d-to-LR*). If the actual journey was R-L, and *d-to-RL* was less than *d-to-LR*, the lap at that location was classified as a correct generalization, and vice versa. The figure shows an example in which a reference subtask is VD and a neuronal trajectory in the R-L journey of an SA subtask was examined. (**B**) The generalization accuracy for determining a R-L/L-R journey from the neuronal trajectory in each lap in two of the three subtasks based on the distance to the mean neuronal trajectories in a remaining reference subtask as a function of the number of cells (red: reference: VD, target: SA and DA; green: reference: SA, target: VD and DA; blue: reference DA, target: VD and SA). Cells recorded from five rats were combined. The accuracy was calculated based on the most frequently classified journeys in the central stem. The error bars show the SD for 1000 random choices of the cells used; the dashed line shows the chance level, and the bar next to the dashed line shows the range of accuracies using random shuffling (control).

obtained 5–10 times in the reward zone. The neuronal recordings during the stimulation period were removed prior to analysis.

## Task and behavioral training

The task demanded a visually or memory-guided decision in VD, SA and DA at a decision point in the figure 8-shaped maze (overall: 100 cm by 140 cm, 20 cm height; path: 20 cm width; matte black acryl resin, *Figure 1A*). To prevent the rats from accessing unnecessary distal room cues, the task was performed under dim light. Initially, two barriers were put into the maze to construct an O-shaped track. After electrode implantation, the rats were trained to run along the left/ right O-shaped track unidirectionally for reward signals (MFB stimulation) at a reward zone (*Figure 1A*, RZ). Once the rats were running smoothly along both left and right O-shaped tracks (~3 to 7 days), all of them were trained to run in the VD subtask for a reward in the figure 8-shaped maze. In the VD subtask, one of the two visual cues (LED-lights at the right and left corners; *Figure 1A*) was illuminated randomly; the visual cue at the decision point of the maze (*Figure 1A*, DP) indicated which direction to turn to receive reward signals. The rats did not access the visual cue until reaching the decision point (*Figure 1A*) because the height of the walls of the maze (20 cm) was much higher than the level of the rat's eyes (~5 cm) and the rats did not stand up during the experiment. The rats were trained for ~3 to 7 days until they achieved a threshold of 80% decisions correct for >20 laps at a constant running speed (>20 cm/s). Next, they were trained to run in the SA subtask for >20 laps. In the SA subtask, both LEDs were illuminated so that the rats could not rely on the visual cue. Instead, the rat had to choose a direction opposite to the previous one. Those alternating behaviors spontaneously occurred and the rats easily achieved the threshold of 80% decisions correct for >20 laps (~1 to 3 days). Finally, the rats were trained to run in the DA subtask for 20 laps. The DA subtask was almost identical to the SA subtask, except that a 5-s delay period was incorporated. During the delay period, a barrier appeared for 5 s, 20 cm ahead of the entrance to the central stem (*Figure 1A*, blue line). Because the rats initially tended to move back from the barrier during the first few laps of the DA subtask, two other barriers were set at both sides of the return rail to prevent such behavior until it ceased. The rats achieved a threshold of 90% decisions correct for >20 laps (1 day).

After the rats correctly performed all three subtasks, they continuously performed all the subtasks within about 1 hr in the following sequence—VD (20 laps), SA (20 laps), VD (10 laps), DA (20 laps), VD (20 laps), SA (20 laps), VD (10 laps), and DA (20 laps)—and were rewarded each time they arrived at the reward zone at the correct return rail. The unit-recordings were made within 2–6 training days after each rat conducted this final sequence of subtasks. The number of possible journeys in the VD subtask was twice that of the other subtasks. Thus, to increase the sample size for the VD subtask for analysis, and to clearly distinguish the behaviorally similar SA and DA subtasks, 10 laps of the VD subtask were interpolated between the SA and DA subtasks. The rats achieved a 95% correct decision rating from the beginning of the continuous task.

## Spike sorting

Spike sorting was performed offline using an automatic spike sorting software program with KlustaKwik (*Harris et al., 2000*) and FastICA (*Hyvarinen, 1999*) called 'ICSort', which is a custom-made program using Matlab and the C++ programming language. ICSort completed two steps: clustering and separation. Initially, in the clustering step, ICSort automatically sorted the first, second and third principal components' feature vectors of all the extracted spikes in the high-dimensional clustering space using KlustaKwik. Then, the extracted spike waveforms in each sorted cluster were concatenated. Next, in the separation step, ICSort used FastICA to find spatial filters to separate the spike waveforms into spatially fixed and distinct, maximally independent components (ICs). To determine which ICs were indicative of neuronal activity, spikes of >50 µV were extracted from each IC and checked to determine whether there was a clear refractory period in the inter-spike interval (ISI) histogram. Finally, the ICSort aggregated these distributed ICs into single units based on each ICA basis vector, which represents the relative distance between the source of the IC and the tip of the electrode. This approach has been previously described in detail (*Takahashi et al., 2003a*, *2003b*; *Takahashi and Sakurai, 2007*, *2009a*, *2009b*).

## Place maps

The spatial firing rate distributions ('place fields') for each cell were constructed in the standard manner by summing the total number of spikes that occurred in a given location bin (5 cm by 5 cm), dividing by the amount of time that the rat spent in that location, and smoothing with a Gaussian centered on each bin (*Leutgeb et al., 2005*). The firing rate at a point, x, was calculated as:

$$f(x) = \sum_{i=1}^{n} w(\frac{s_i - x}{h}) \Big/ \int_0^T w(\frac{y(t) - x}{h}) dt,$$

where $n$ is the number of spikes, $s_i$ the position of the $i$-th spike, $y(t)$ the position of the rat at time $t$, and (0, T) the period of the recording. The kernel function, $w$, was a Gaussian of $h$ = 5 cm width. Positions more than 5 cm away from the tracked path were regarded as unvisited.

## Cell identification and recording periods

After spike sorting, putative pyramidal cells were distinguished from putative fast spiking interneurons by the spike width (0.4 ms) and the average firing rate (5 Hz). The low rate cells (<0.1 Hz) were excluded. Only cells identified as pyramidal were used in the analyses outlined below.

It is well known that during immobility periods, including delay periods, hippocampal activity is not place-specific (*McNaughton et al., 1983*; *Pastalkova et al., 2008*). Thus, spikes during immobility periods in the maze were excluded.

## Spatial information

The spatial information was calculated as the number of bits per spike according to the formula (*Skaggs et al., 1993*):

$$Spatial\,Information = \sum_i P_i (R_i/R) \log_2 (R_i/R),$$

where $i$ indexes over the position bins in the trial type, $P_i$ is the probability that the rat was in bin $i$, $R_i$ the mean firing rate in bin $i$, and $R$ the overall mean firing rate over the trial type. More spatially specific firing leads to a larger value for this measurement.

## Statistical comparison of firing rates using ANCOVA

To analyze the activity of cells in the central stem of the maze, the central stem was divided into spatial bins of equal length (approximately 0.35 cm). The following parameters were calculated for each traversal through each of the bins: (1) firing rate was calculated as the spike count in the bin divided by the time spent in the bin, and convolved with the Gaussian (SD = 5 cm); (2) running speed was measured as the time to traverse the constant length bin; (3) head direction was calculated as the mean of the animal's head directions (in degrees) within the bin; (4) lateral position was represented by the mean of the x coordinates of the rat's head within the bin. For each cell, a one-way ANCOVA was performed between different journeys within a given subtask or between different task-demands within a journey as independent (fixed) variables, firing rate as the dependent measure, and with running speed, head direction and lateral position as covariates (R version 2.15.2; The R Project for Statistical Computing, Vienna, Austria).

## Spatial similarity

The spatial similarity of the place fields between two trial types or repeated exposures was calculated using the spatial correlation between the maps (*Leutgeb et al., 2005*). The region of interest (ROI) is defined along a common running route (i.e., the central stem; *Figure 4A*) between the trial types or sessions.

## Rate similarity

Changes in the firing rates between two trial types were expressed by calculating a modified difference/sum score (*Leutgeb et al., 2005*). The score for a pair of trial types or repeated exposures was obtained by calculating the unsigned difference between the rates in the two trial types or repeated exposures and dividing the difference by the sum of the two rates. To maintain consistency across spatial similarity, the rate similarity was obtained by subtracting the score from one. The possible scores ranged from an asymptotic value of 0–1. The ROI is defined in the same way as spatial similarity.

## Neuronal trajectory analysis

The dynamics of the neuronal ensemble activity were analyzed as a trajectory through a state space (neuronal trajectory; *Figure 5*) (*Briggman et al., 2005*; *Mazor and Laurent, 2005*; *Harvey et al., 2012*). Initially, the path of the rat was linearized for each journey by projecting the actual trajectory followed by the rat on that lap onto a user-defined idealized path using nearest neighbor Delaunay triangulation. Spatial bins had a resolution of approximately 0.35 cm. At each location on the linearized path, the spike count that convolved with the Gaussian (SD = 5 cm) of the ensemble containing $n$ simultaneously monitored cells was defined as a point in an $n$-dimensional space, with each dimension representing the activity of a single neuron. Since the neuronal trajectory is in $n$-dimensional space, to visualize the neuronal trajectory in three-dimensional space, principal component analysis (PCA) was used as a dimension-reduction technique. The data were organized into $m$ locations by an $n$-cell data set.

The distances between the trajectories were measured as Euclidian distances between the corresponding locations. The classification was performed for individual laps based on the distances to the mean lap trajectories. If the distance to the mean target trajectory was less than the distance to the other mean trajectories, the location for that lap was classified as the targeted choice. Classification accuracy was calculated at each location by averaging across laps for the individual trial types. In all of the distance classifications, the mean trajectories were calculated excluding the test trajectory (i.e., leave-one-out cross validation). In *Figure 8*, the classification accuracy was measured for each rat based on the most frequently classified choices at the central stem of the maze. The target cells were all simultaneously monitored place cells in each rat. In *Figure 9*, the classification accuracy for generalization was calculated based on the most frequently classified choices at the central stem of the maze; the combination of target cells for the classification were randomly chosen 1000 times. For control conditions, shuffling was performed by randomizing the sequence of laps 1000 times.

## Behavior analysis

The rats could make a decision as soon as the visual cue was detected. However, the exact point where the rats mentally detect the visual cue cannot be estimated because measuring the awareness of the

visual cue for rats is difficult to perform. Instead, the timing of the behavioral output was estimated based on the location of turn onset. The turn onset was defined as the last location when the rat was in the central stem or the upper junction of the maze (i.e., DP in *Figure 1A*) and when the angular velocity was below a threshold value (0.12 rev/s) (*Harvey et al., 2012*) (*Figure 1C*). These locations of turn onset were considered the signature of the rats' behavioral decision in the analyses.

## Statistical analysis

The spatial and rate similarities, behavior analysis and neuronal trajectory analysis were performed using custom-made programs based on Matlab functions (Matlab 7.14; MathWorks, Natic, MA). All comparisons for the spatial and rate similarities were completed using the Wilcoxon rank sum test. The statistical test for classifier accuracy was performed using the binomial test. The statistical ANCOVA comparison was performed using Matlab functions in conjunction with R (R version 2.15.2; The R Project for Statistical Computing).

## Histology

After the task was performed, the rats were sacrificed by pentobarbital sodium overdose and perfused with formalin. The brains were cut coronally at 30 μm and stained with cresyl violet. Each section through the relevant part of the hippocampus was retained for analysis. All dodecatrodes were identified and the tip of each electrode was found by comparing it with the adjacent sections. The recordings from a given dodecatrode were included in the data analysis if its deepest position was under the CA1 pyramidal cell layer.

## Acknowledgements

I would like to thank Y. Sakurai for the initial design of the behavioral task and for discussion and comments on the manuscript, all the members of the Sakurai and Fujiyama laboratories for technical assistance, and M. Kawato and all of the advisors and members of the JST PRESTO program 'Decoding and controlling brain information' for their fruitful comments and encouragement.

## Additional information

### Funding

| Funder | Grant reference number | Author |
| --- | --- | --- |
| JST PRESTO | | Susumu Takahashi |
| JSPS KAKENHI | 23650158, 24300148 | Susumu Takahashi |

The funders had no role in study design, data collection and interpretation, or the decision to submit the work for publication.

### Author contributions

ST, Designed the tasks, set up the behaviour and electrophysiology apparatus, performed the surgery, behaviour and electrophysiology experiments, data collection and analyses, and wrote the paper.

### Ethics

Animal experimentation: All procedures were approved by the Doshisha University (approved number: 1229) and Kyoto Sangyo University (approved numbers: 2011-04 and 2011-05) Institutional Animal Care and Use Committees.

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
