## [Decision Letter]

Thank you for choosing to send your work entitled “Hierarchical organization of context in the hippocampal episodic code” for consideration at *eLife*. Your article has been evaluated by a Senior editor and 3 reviewers, one of whom is a member of our Board of Reviewing Editors.

The following individuals responsible for the peer review of your submission want to reveal their identity: Howard Eichenbaum (Reviewing editor), Jozsef Csicsvari (peer reviewer), and Emma Wood (peer reviewer).

The Reviewing editor and the other reviewers discussed their comments before we reached this decision, and the Reviewing editor has assembled the following comments based on the reviewers' reports.

* The reviewers agree that the paper could be useful and important, but requires additional analyses and improved presentation of the findings. There is a major concern about the need to control for potential differences between location, speed, and head direction in making claims about differential firing on the stem of the maze prior to left and right turn trials (see, for example, Wood et al., 2000). This will require substantial further analyses and likely restriction of the analyses to the central portion of the maze stem.

* There are also concerns about the language and presentation of data. Part of the problem is the use of too much jargon.

* In addition, the description of generalization is difficult to follow. In particular, Figure 7 is hard to interpret, so perhaps the data can be presented in a more intuitive way. The premise is understandable – that knowing the R-L and L-R neuronal trajectories in one task allows classification into R-L and L-R for the other tasks, but the figure does not aid that understanding. Also, it would be useful that provide some analysis and discussion of the relative ability to classify into trajectories (L-R vs R-L) vs into tasks (VD, SA, DA) vs into specific trial types (i.e., task and trajectory information combined).

[Editors' note: the following comments were also sent to the author before acceptance.]

The author attempted to address the concerns of all the reviewers by performing ANCOVAs to eliminate cells for which differences in direction, speed, or lateral position could account for differences in firing patterns associated with distinct trajectories. However, nearly all the examples are taken from areas of the maze stem at the beginning or end of the stem where the animal's behavior, location, and direction of movement clearly differ. This is evident in the nearly all the examples shown in Figure 3 (exceptions are RatF120112-cell 53 and RatG111125-cell 326). We fear that somehow the ANCOVAs are not detecting this confound and this could undermine the conclusions. We recommend that you select a middle region of the maze stem that excludes the top and bottom areas where the trajectory of the rat separates, and use only the data from the middle region for all analyses.

---

## [Author Response]

** The reviewers agree that the paper could be useful and important, but requires additional analyses and improved presentation of the findings*.

The entire manuscript has been thoroughly rewritten and reconstructed. According to the reviewers' suggestions, two additional analyses were performed. To improve the presentation of the findings, some comments and figures for explanation were added.

*There is a major concern about the need to control for potential differences between location, speed, and head direction in making claims about differential firing on the stem of the maze prior to left and right turn trials (see, for example, Wood et al., 2000). This will require substantial further analyses and likely restriction of the analyses to the central portion of the maze stem*.

In the revised manuscript, I performed an additional analysis using ANCOVA as Wood et al., 2000 had conducted. The results are shown in Figure 4 of the revised manuscript and suggest that the differential firing in the central stem of the maze was significant even when running speed, head direction, and lateral position are taken into account. This is consistent with the findings of Wood et al., 2000. I added a paragraph describing this to the Results section and a few sentences to the Discussion section. To describe my methods in detail, I added a paragraph to the Materials and methods section.

** There are also concerns about the language and presentation of data. Part of the problem is the use of too much jargon*.

In line with the reviewers' suggestions, I have thoroughly rewritten and reconstructed the previous manuscript. In particular, I have replaced the terms “spatiotemporal context”, “non-spatial context”, and “combination of spatiotemporal and non-spatial contexts” with “journey”, “task-demand”, and “trial type”, respectively, throughout the Results, Discussion, and Materials and methods sections, and the figure legends in the revised version of the manuscript. In addition, I have replaced “spatial measure” with “spatial correlation”.

** In addition, the description of generalization is difficult to follow. In particular, Figure 7 is hard to interpret, so perhaps the data can be presented in a more intuitive way. The premise is understandable – that knowing the R-L and L-R neuronal trajectories in one task allows classification into R-L and L-R for the other tasks, but the figure does not aid that understanding*.

To clearly demonstrate the presentation of the data for generalization accuracy, I have added a subfigure to Figure 9 (Figure 7 in the previous version of the manuscript) and some comments to the legend of Figure 9. In addition, to clearly describe the generalization analysis, I have added a few sentences to the Results section.

*Also, it would be useful that provide some analysis and discussion of the relative ability to classify into trajectories (L-R vs R-L) versus into tasks (VD, SA, DA) versus into specific trial types (i.e., task and trajectory information combined)*.

In line with the reviewers' suggestions, I have performed an additional analysis to assess the relative ability to classify journeys, versus task-demands, versus specific trial types using a binary classification scheme. Figure 8 of the revised manuscript shows these results. The results suggest that the abilities of the hippocampal ensemble activity to identify journeys, task-demands, and trial types are sufficiently high and relatively similar to each other. I added details pertaining to my methods to the Materials and methods section. Moreover, I have added some text addressing these results to the Results and Discussion sections.

*[Editors' note: the following comments were also sent to the author before acceptance.*]

*The author attempted to address the concerns of all the reviewers by performing ANCOVAs to eliminate cells for which differences in direction, speed, or lateral position could account for differences in firing patterns associated with distinct trajectories. However, nearly all the examples are taken from areas of the maze stem at the beginning or end of the stem where the animal's behavior, location, and direction of movement clearly differ. This is evident in the nearly all the examples shown in Figure 3 (exceptions are RatF120112-cell 53 and RatG111125-cell 326). We fear that somehow the ANCOVAs are not detecting this confound and this could undermine the conclusions*.

The reviewers pointed out that the detection error of the ANCOVAs could undermine the main conclusions because many place fields in Figure 3 in the previous manuscript can be seen at the beginning or end of the stem. I have addressed this concern in three ways.

1) **Results of the reanalysis**: I completely agree with the reviewers' concern about Figure 3. However, although I repeatedly checked place fields in Figure 3 in the previous manuscript, the results suggested that they fired at significantly different rates in the central stem even when head direction, running speed, and lateral position in the central stem were taken into account. For example, although RAT# G111125 Cell 109 in Figure 3b in the previous manuscript showed a large part of the place fields at the beginning or end of the stem (dotted red box in Author response image 1), all analyses in the previous manuscript excluded such regions. Instead, only other remaining parts of the place fields in the central stem (i.e., locations where the rat's head direction, running speed, and lateral position do not clearly differ) were analyzed in the previous manuscript (solid red box in Author response image 1; arrows: analyzed place fields) and showed significant rate differences. Therefore, I concluded that there is no serious concern about the detection error of the ANCOVAs.Author response image 1.Red solid and dotted boxes indicate the analyzed and unanalyzed regions of the central stem, respectively.Arrows indicate the location of the analyzed place fields. The rate was coded on a color scale from blue (silent) to red (maximum rate). The pixels that were not sampled are white. The symbols above the map indicate the journey, task-demand and illuminated light-cue. For each cell, the rate scale corresponds to the peak firing rate in Hz in that condition (indicated to the right of the rate maps).

2) **The bias of the place fields location shown in Figure 3**: As the journey-dependent coding was changed in a global remapping manner, if the place fields of a place cell in one of two journeys were located at the beginning or end of the central stem, the fields in another journey, if any, tended to be located at the another side (e.g., Author response Image 2, RAT# G111125 Cell 109 in Figure 3b in the previous manuscript). The length of the central stem was 60cm and the place field width was approximately 20–30cm so that the beginning and end of the central stem tended to be exclusive to each other. In addition to the tendency, to demonstrate the place fields in the central stem in both left-to-right and right-to-left journeys, and the main finding (i.e., the journey-dependent coding was changed in a global remapping manner), the place fields in Figure 3 in the previous manuscript were selected. This could have caused the reviewers' concern.Author response image 2.The journey-dependent global remapping may make the place fields be exclusively located at the beginning or end of the central stem.The rate was coded on a color scale from blue (silent) to red (maximum rate). The pixels that were not sampled are white. The symbols above the map indicate the journey, task-demand, and illuminated light-cue. For each cell, the rate scale corresponds to the peak firing rate in Hz in that condition (indicated to the right of the rate maps).

*We recommend that you select a middle region of the maze stem that excludes the top and bottom areas where the trajectory of the rat separates, and use only the data from the middle region for all analyses*.

3) **Analyses of the place fields in the middle region of the central stem**: In line with the reviewers' suggestion, I selected cells which showed the place fields restricted in a middle region of the central stem and fired at significantly different rates between different journeys or between different subtasks even when running speed, head direction, and lateral position were taken into account. The results shown in Author response images 4–9 did not undermine the main conclusions, with the exceptions that only the distributions of the change in firing rates in the place fields between VD3 and SA2 were similar to control condition (n.s., *P*=0.44, Wilcoxon rank sum test, Author response image 5), and that the relative ability of distinguishing trial types was slightly higher than that of distinguishing task-demands (*P*=0.03, Kruskal-Wallis test with Tukey honestly significant difference (HSD) post-hoc test, Author response image 8). However, as the number of selected cells for each ensemble (K: 3 cells, F: 4 cells, L: 9 cells, B: 16 cells, G: 27 cells; Only 5% (59/1119) of the recorded cells) was too few to statistically analyze at the ensemble level, these minor exceptions might be due to the sampling bias.

Taken together, as I concluded that the reason that nearly all the examples in Figure 3b are taken from areas of the maze stem at the beginning or end of the stem was not due to the detection error of the ANCOVAs, but due to the selection bias influenced by the global remapping of place fields, I would instead fear that the restricted selection of cells in the middle of the central stem causes the sampling bias and could undermine the main conclusions. Therefore, the data in this response letter were not included in the revised manuscript.

*Also, in all examples, indicate that middle region on illustrations show only examples of cells that differ in firing pattern in that middle section*.

As I totally agree with the reviewers' concern about Figure 3b, I removed ambiguous cells and replaced five cells in Figure 3b in the previous manuscript with cells that fired at significantly different rates in the middle of the central stem in the revised manuscript (see Author response image 3; boxes indicate the middle of the central stem).Author response image 3.Journey-dependent coding of a place-specific activity modulated with task-demand.The rate maps for five CA1 pyramidal cells. The maps are averages for repeated sessions. Each row shows the data from one pyramidal cell. Note that different journeys in the same place resulted in place fields similar in locations, but different task-demands caused different firing intensities in those place fields. Boxes indicate the middle region of the central stem. The rate was coded on a color scale from blue (silent) to red (maximum rate). The pixels that were not sampled are white. The symbols above the map indicate the journey, task-demand, and illuminated light-cue. For each cell, the rate scale corresponds to the peak firing rate in Hz in that condition (indicated to the right of the rate maps).Author response image 4.Proportion of place cells that fired at significantly different rates in the middle region of the central stem.Author response image 5.A quantitative assessment of the spatial and rate similarities between a pair of trial types for place cells that fired at significantly different rates in the middle region of the central stem.(a) The region of interest (ROI) was set at the region highlighted in the green shaded box. (c) In contrast to figure 4, only the distributions of the change in firing rates in the place fields between VD3 and SA2 were similar to control condition (n.s., *P*=0.44, Wilcoxon rank sum test).Author response image 6.Neuronal trajectory of different journeys and different future directions for place cells that fired at significantly different rates in the middle region of the central stem.Author response image 7.Episode-specific neuronal trajectories for place cells that fired at significantly different rates in the middle region of the central stem.Author response image 8.Comparison of accuracy of distinguishing journeys, task-demands, and trial types from neuronal ensemble activity of place cells that fired at significantly different rates in the middle region of the central stem.In contrast to figure 8, the average classification accuracy of distinguishing trial types was different from that of distinguishing task-demands (*P*=0.03, Kruskal-Wallis test with Tukey honestly significant difference (HSD) post-hoc test).Author response image 9.Generalization accuracy of determining a journey over task-demands from neuronal ensemble activity of place cells that fired at significantly different rates in the middle region of the central stem.